# Podocyte-specific knockout of the neonatal Fc receptor (FcRn) results in differential protection depending on the model of glomerulonephritis

**James F. Dylewski** [1,2]*, **Pantipa Tonsawan**[3], **Gabriela Garcia**[1], **Linda Lewis**[1], **Judith Blaine**[1]

**1** Division of Renal Disease and Hypertension, University of Colorado School of Medicine, Aurora, CO, United States of America, **2** Department of Nephrology, Denver Health Medical Center, Denver, CO, United States of America, **3** Division of Nephrology, Khon Kaen University, Khon Kaen, Thailand

* James.Dylewski@cuanschutz.edu

**Data Availability Statement:** All relevant data are within the manuscript and its Supporting Information files.

## Abstract

Podocytes have been proposed to be antigen presenting cells (APCs). In traditional APCs, the neonatal Fc receptor (FcRn) is required for antigen presentation and global knockout of FcRn protects against glomerulonephritis. Since podocytes express FcRn, we sought to determine whether the absence of podocyte FcRn ameliorates immune-mediated disease. We examined MHCII and costimulatory markers expression in cultured wild type (WT) and FcRn knockout (KO) podocytes. Interferon gamma (IFNγ) induced MHCII expression in both WT and KO podocytes but did not change CD80 expression. Neither WT nor KO expressed CD86 or inducible costimulatory ligand (ICOSL) at baseline or with IFNγ. Using an antigen presentation assay, WT podocytes but not KO treated with immune complexes induced a modest increase in IL-2. Induction of the anti-glomerular basement membrane (anti-GBM) model resulted in a significant decrease in glomerular crescents in podocyte-specific FcRn knockout mouse (podFcRn KO) versus controls but the overall percentage of crescents was low. To examine the effects of the podocyte-specific FcRn knockout in a model with a longer autologous phase, we used the nephrotoxic serum nephritis (NTS) model. We found that the podFcRn KO mice had significantly reduced crescent formation and glomerulosclerosis compared to control mice. This study demonstrates that lack of podocyte FcRn is protective in immune mediated kidney disease that is dependent on an autologous phase. This study also highlights the difference between the anti-GBM model and NTS model of disease.

## Introduction

Despite glomerulonephritis being the third leading cause of end stage kidney disease in the United States [1], treatment options are limited and typically involve systemic immunosuppressive medications which are variably effective and have multiple side effects. Treatment of

**Funding:** This work was supported by the National Institute of Diabetes and Digestive and Kidney Disease (NIDDK) [grants 1R01DK104264-01A1, 5R01DK082509-05, and 5T32DK007135-42] a Norman Coplon Satellite Healthcare grant, and a Denver Health Medical Center Pilot Grant.

**Competing interests:** The authors have declared that no competing interests exist.

these diseases is limited, in part, by the incomplete understanding of how the immune system recognizes and targets the kidneys.

Prior studies have demonstrated that glomerulonephritis are dependent on CD4[+] T cell and the adaptive immune system [2–4]. Stimulation of the CD4[+] T cells have been shown to be carried out by intrinsic renal cells that express major histocompatibility complex class II (MHCII) [5]. Goldwich et al published data showing that podocytes express MHCII and suggesting that podocytes act as non-hematopoietic professional antigen presenting cells (APCs) that can stimulate CD4[+] T cells [6]. Podocytes are able to express CD80, which is a well-known costimulatory marker needed for activation of T cells, and CD80 expression has been associated with certain glomerular diseases [7, 8]. Given these findings, podocytes have been proposed as candidates for the intrinsic renal cells that trigger an immune response and lead to glomerulonephritis. However, a better understanding of the mechanism by which podocytes can cause an immune response is needed to provide targeted therapies for these devastating disorders.

A promising area of study is the neonatal Fc receptor (FcRn). FcRn is a major histocompatibility class I-like protein that was initially described and discovered as a means for infant enterocytes to obtain passive immunity through breast milk [9, 10]. Since its initial description, FcRn has been found to play important roles in albumin and IgG recycling and has been shown to be expressed in several other cell types [9, 11–18]. In the kidneys, FcRn is expressed in endothelial cells, podocytes and proximal tubular cells [13–15, 17, 19].

FcRn also plays an important role in adaptive immunity [12, 20–24]. Studies in dendritic cells have established that FcRn is necessary for trafficking antigen-antibody complexes for degradation as well as presentation on MHCII [10, 20, 25, 26]. Furthermore, studies have shown that when FcRn is absent, dendritic cells cannot present antigen for antigen presentation [25]. Interestingly, when FcRn is globally knocked out of mice, glomerulonephritis is attenuated [27]. However, it remains unclear whether lack of FcRn in dendritic cells, endothelial cells, or podocytes provides this protective effect. Knowledge of precisely in which cells FcRn is required for induction of glomerulonephritis would allow for the development of targeted therapies.

In this study, we investigated whether podocytes can function as APCs and whether knockout of FcRn specifically in podocytes attenuates the progression of anti-glomerular basement membrane (anti-GBM) nephritis and nephrotoxic serum nephritis (NTS). We chose these models as both are well-characterized models of immune mediated kidney disease. In addition, using the anti-GBM model, Goldwich et al. had previously found that podocyte-specific knockout of MHCII attenuated renal disease but whether this can be observed in other disease models and the exact mechanism is unknown [28]. Since in dendritic cells, FcRn is required for antigen presentation via MHCII, we hypothesized podocyte-specific knockout of FcRn would ameliorate antibody-mediated immune kidney disease.

The anti-GBM model is characterized by a heterologous phase which lasts 4–5 days post anti-GBM antibody injection and is characterized by neutrophil invasion and complement deposition [29, 30]. The initial heterologous phase is followed by an autologous phase in which T helper cells enter the kidney in response to autologous antibodies produced to the anti-GBM antibody. In the anti-GBM model, the autologous antibody phase only lasts 4–6 days as the anti-GBM model is a severe model of disease with mice not surviving beyond 10 days [31, 32]. We examined the effects of podocyte-specific FcRn KO at both day 3 and day 8 after induction of anti-GBM nephritis to determine whether lack of FcRn had an effect on either phase of the disease.

The NTS model has relatively mild heterologous phase and is more dependent on autologous antibody production [33, 34]. Since this model is more dependent on the humoral

immune system and takes longer to develop, we examined the effects of the podocyte-specific FcRn KO at 28 days after induction to ensure adequate time for humoral response.

## Methods

### Cell culture

**Generation of conditionally immortalized WT and FcRn KO podocytes.** Podocytes were isolated from WT or global FcRn KO mice and then underwent immortalization using a thermosensitive polyomavirus simian virus 40 (SV40) T antigen as previously described [35–37]. Primary podocytes were allowed to replicate at 33°C. To induce differentiation, podocytes were placed at 37°C for 8–10 days. To verify expression of podocyte markers, podocytes were stained with podocin or WT1 and synaptopodin expression was checked by Western blot. Conditionally immortalized WT or KO podocytes were allowed to differentiate at 37°C for 9–10 days prior to being used in experiments.

### Flow cytometry

Multiparameter flow cytometry (FACS) was performed using a Beckman Coulter Gallios 561 compensated with single fluorochromes and analyzed using Kaluza™ software (Beckman Coulter). Differentiated transformed mouse WT and FcRn KO podocytes populations were identified by their characteristic forward scatter/side scatter (fsc:ssc) properties. Monoclonal anti-mouse antibodies anti-CD80-FITC (clone 16-10A1), anti-MHCII-APC (clone MS/114.15.2), anti-CD86-PE-Cy5 (clone GL1) were purchased from eBiosciences and anti-PE-ICOSL (clone 9F.8A4) was purchased from BioLegend. The Live/Dead Fixable Aqua Dead Cell Stain Kit (Invitrogen) was used to assess the number of live cells. Cells were detached, washed and stained for surface antigen expression at 4°C in the dark for 60 minutes and then washed again prior to flow cytometry. Unstained cells and fluorescence minus one (FMO) samples were used as controls.

### Antigen presentation assay

$1 \times 10^3$ WT or FcRn KO podocytes per well were plated in a collagen coated 96 well plate and allowed to differentiate. After differentiation, podocytes were treated for 24 hours with IFNγ (100 units/ml) media. Subsequently, podocytes were treated for 6 hours with media alone, an 4-Hydoxy-3-nitrophenylacetyl-ovalbumin (NP-ovalbumin) specific mouse IgG2c antibody (kindly provided by Raul Torres, PhD, University of Colorado Denver) alone at a concentration of 20 μg/ml or immune complexes made by incubating mouse IgG2a antibody (20 μg/ml) with NP-ovalbumin (40 μg/ml). After treatment, podocytes were rinsed well. The NP-ovalbumin specific CD4+ hybridoma cell line BO80.1 (kindly provided by Philippa Marrack, PhD, National Jewish) was added to each well at a density of $1 \times 10^4$ for 24 hrs. The plate was then spun down and supernatants transferred to a clean 96 well plate and frozen to remove any residual T cells. IL-2 concentration in the supernatant was measured by ELISA. Each experimental condition was repeated in triplicate and an n of at least 4 experiments was performed.

### Animals

podFcRn KO mice were obtained by crossing C57BL/6J FcRn floxed mice [38] (a kind gift from Dr. Sally Ward, UT Southwestern) with C57BL/6J podocin-Cre mice (Jackson Labs, Bar Harbor, Maine) as previously described [36]. Genotype was determined by PCR. All experimental mice were homozygous for the floxed (fl) FcRn gene. podFcRn KO mice (FcRn fl/fl; Pod-cre/+) were double transgenic resulting in no FcRn expression in podocytes. Control

mice (FcRn fl/fl;+/+) were single transgenic (no Cre expression) resulting in unchanged FcRn expression in podocytes. Male mice were used for all experiments. Euthanasia was achieved by euthanasia solution overdose. All procedures involving animals were performed using protocols approved by the Institutional Animal Care and Use Committee at the University of Colorado, Denver. All experiments were performed in accordance with regulations and policies as laid out by PHS Policy on Humane Care and Use of Laboratory Animals as well as the Institutional Animal Care and Use Committee at the University of Colorado, Denver.

For the anti-GBM experiments mortality was as follows: one control and two podocyte-specific FcRn Ko mice died during the experiment. Starting numbers were 11 control and 12 podocyte-specific FcRn KO mice.

For the NTS experiments, one control and one podocyte-specific FcRn KO mouse died during the experiment so starting numbers were as follows: 9 control and 8 podocyte-specific FcRn KO mice

## Anti-glomerular basement membrane nephritis model

The anti-GBM nephritis model was induced as previously described [29–31]. Briefly, 8–-12-week-old podFcRn KO or control mice were primed with 1 mg/mouse of normal rabbit IgG in Freund's complete adjuvant, followed 5 days later by an intravenous injection of 0.2 mg/g of rabbit anti-mouse GBM antibody provided by Dr. Gabriela Garcia. Mice were sacrificed at 8 days after GBM antibody injection. Prior to sacrifice urine and blood were collected. Urine albumin was measured using the Albuwell assay (Exocell), urine creatinine was measured using the assay and BUN was measured on an Alpha Wasserman auto analyzer. Titers of rabbit IgG in control and podFcRn KO mice were measured as previously described [39]. Briefly, a 96 well plate was coated with 10 ug/ml rabbit IgG. Coated wells were incubated with serum from control or podFcRn KO mice diluted 1:10,000. Mouse anti-rabbit total IgG titers were measured by ELISA (Biolegend).

## Nephrotoxic serum nephritis model

The nephrotoxic serum nephritis model is a newer but increasingly utilized model of glomerulonephritis in mice [33, 40–45]. Briefly, 8–12 week old podFcRn KO or control mice were given an intravenous injection of 100μL of sheep anti-Rat glomerular basement membrane antibody made by Probetex Inc. (San Antonio, TX) at day 0. Mice were sacrificed on day 28 after injection. Prior to sacrifice, urine and blood were collected. Urine albumin, urine creatinine, and serum BUN were measured using the same methods as described above under the anti-GBM nephritis model.

## Histology

Three micrometer (μm) sections were cut from paraffin embedded tissue and stained using the periodic acid Schiff reagent. All histologic analysis was performed in a blinded fashion. Crescent formation, defined as the presence of two or more layers of cells in Bowman's space, was evaluated in 20–30 random glomeruli for each mouse. Glomerulosclerosis was evaluated using a semi-quantitative scoring system. The glomerular score was determined from a mean of at least 20–30 glomeruli sampled at random. The severity of glomerulosclerosis was graded on a scale (0 to 4) as follows: grade 0, normal glomerulus; grade 1,mild hyalinosis/sclerosis involving < 25% of the glomerulus; grade 2, moderate segmental hyalinosis/sclerosis involving < 50% of the glomerulus; grade 3, diffuse glomerular hyalinosis/sclerosis involving more than 50% of the glomerulus; grade 4, diffuse glomerulosclerosis with total glomerular obliteration. Histologic analysis was performed using Aperioscope.

## Immunofluorescence

Confocal microscopy images were acquired using Zeiss 780 laser-scanning confocal/multiphoton-excitation fluorescence microscope with a 34-Channel GaAsP QUASAR Detection Unit and non-descanned detectors for two-photon fluorescence (Zeiss, Thornwood, NY). The imaging settings were initially defined empirically to maximize the signal-to-noise ratio and to avoid saturation. In comparative imaging, the settings were kept constant between samples. Images were captured with a Zeiss C-Apochromat 40x/1.2NA Korr FCS M27 water-immersion lens objective. The illumination for imaging was provided by a 30mW Argon Laser using excitation at 488 nm, Helium Neon (HeNe) 5mW (633 nm) and 1mW (543 nm). Image processing was performed using Zeiss ZEN 2012 software. Images were analyzed in Image J software (NIH, Bethesda, Maryland).

For the in vivo immunolocalization studies, the kidneys were cleared of blood by perfusion of PBS and then with 4% PFA in PBS (pH 7.4). The kidneys were then removed, immersed in 4% PFA for 24hr, infused with 5% (2 hr), 10% (2 hr) and 25% (overnight) sucrose, frozen in liquid nitrogen and cryosectioned (3 μm). Kidney sections were blocked (10% normal goat serum and 1% bovine serum albumin (BSA) in PBS) and incubated overnight at 4˚C with primary antibody ((Ly6G, 1:10; BD/Pharmingen), FITC-C3 (1:200; MP Biomedical), CD68 (1:200; Bio-Rad), CD4 (1:50; Biolegend). After washing, the sections were incubated (60 min, room temperature) with appropriate mix of Alexa 488-conjugated goat anti-chicken IgG (1:500; Invitrogen), Hoechst 33342 (1:1000; ThermoFisher) and Alexa 633-conjugated phalloidin (1:250; Invitrogen). Sections were then washed with PBS and mounted in VectaShield (Vector Labs). Fluorescence intensity was quantified using ImageJ. Number of macrophages or CD4+ T cells were determined by counting nuclei, identified by Hoechst, that were also positive for CD68 or CD4 staining.

## Data analysis

Data are presented as means ± SE. Statistical analysis was performed using *t*-tests for two groups and one-way analysis of variance for 3 or more groups, using Prism software (Graph-Pad, San Diego, CA). Tukey's post hoc test was applied to the ANOVA data. Values were considered statistically significant when $p < 0.05$.

## Results

### MHCII and costimulatory marker expression in WT and FcRn KO podocytes

Podocytes were isolated from wild type (WT) and global FcRn KO (KO) mice and immortalized using the thermosensitive polyomavirus simian virus (SV40) T antigen as described previously [36, 37]. Murine podocytes are not known to express MHCII at baseline. However, it is well established that cells, including podocytes, can express MHCII after exposure to INFγ [6, 46–52]. In order to determine if the absence of FcRn alters podocytes ability to express MHCII or costimulatory markers which, in turn, could affect antigen presentation, WT and KO podocytes underwent flow cytometry for MHCII, cluster of differentiation 80 (CD80), cluster of differentiation 86 (CD86) or inducible costimulatory ligand (ICOSL) expression at baseline and after treatment of IFNγ.

Both WT and KO podocytes had minimal MHCII expression at baseline. After treatment with IFNγ, both WT and KO had a significant increase in MHCII expression compared to untreated cells. (26.5% +/- 7.6% for WT + IFNγ vs. 23.9% +/- 2.6% for KO + IFNγ, Fig 1).

Both WT and KO podocytes expressed cluster of differentiation 80 (CD80) at baseline (25.8 +/- 2.1% for WT versus 26.8 +/- 3.2% for KO) and expression was not significantly changed

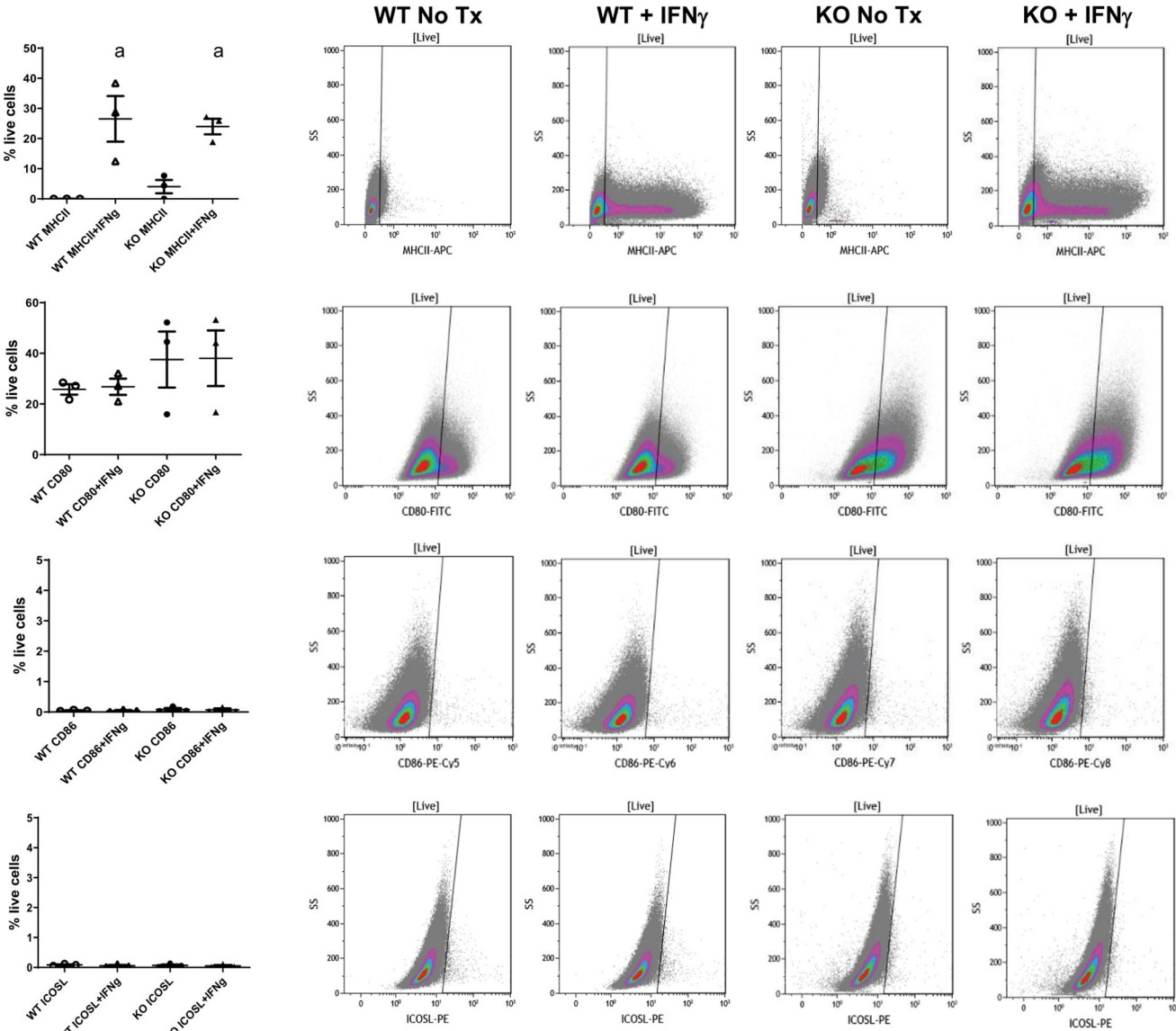

**Fig 1. MCH II and co-stimulatory marker expression in cell lines in WT and FcRn KO podocytes.** Flow cytometry data showing MHC II, CD80, CD86 and ICOSL expression in WT and FcRn KO podocytes at baseline and after treatment with IFNγ. Treatment with IFNγ results in increased MHCII expression in WT and KO podocytes (a, p = 0.0036). Both WT and FcRn KO podocytes express CD80 at baseline and this is not significantly increased after treatment with IFNγ. WT and FcRn KO podocytes do not express CD86 or ICOSL at baseline or after treatment with IFNγ (n = 3 to 4 experiments for all conditions).

with IFNγ treatment (Fig 1B). Both WT and KO podocytes did not express cluster of differentiation 86 (CD86) or inducible costimulatory ligand (ICOSL) at baseline or after treatment with IFNγ (Fig 1).

### WT podocytes were weak antigen presenting cells in vitro whereas KO podocytes did not present antigen at all

In addition to T cells binding to the APC's MHCII and costimulatory markers, CD4+ T cells need autocrine cytokine production (such as interleukin 2 (IL2)) to act as a third signal for T cell activation. To determine whether podocytes can act as APCs, an in vitro

antigen presentation assay was performed using WT and KO podocytes. CD4[+] T cell activation was measured by determining the amount of IL2 produced by the T cells when WT or KO podocytes were used as APCs. WT podocytes treated with media alone and immunoglobulin G (IgG) alone induced minimal IL2 production when co-cultured with CD4[+] T cells. When WT podocytes were treated with IgG + ovalbumin (immune complexes, ICs) there was a significant increase in IL2 production by CD4[+] T cells (6.8 +/- 0.9 pg/ml versus 3.3 +/- 0.7 pg/ml for WT + IC vs. WT + media, $p < 0.05$, Fig 2). T cell production of IL2, however, was significantly less when WT podocytes were used as APCs compared to splenocytes (S1 Fig), suggesting that podocytes are inefficient APCs. KO podocytes treated with media or IgG alone also induced minimal IL2 production by CD4[+] T cells. In contrast to WT podocytes, when FcRn KO podocytes were used as APCs, there was no significant increase in IL2 production by CD4[+] T cells (Fig 2), suggesting that KO podocytes are unable to present antigen.

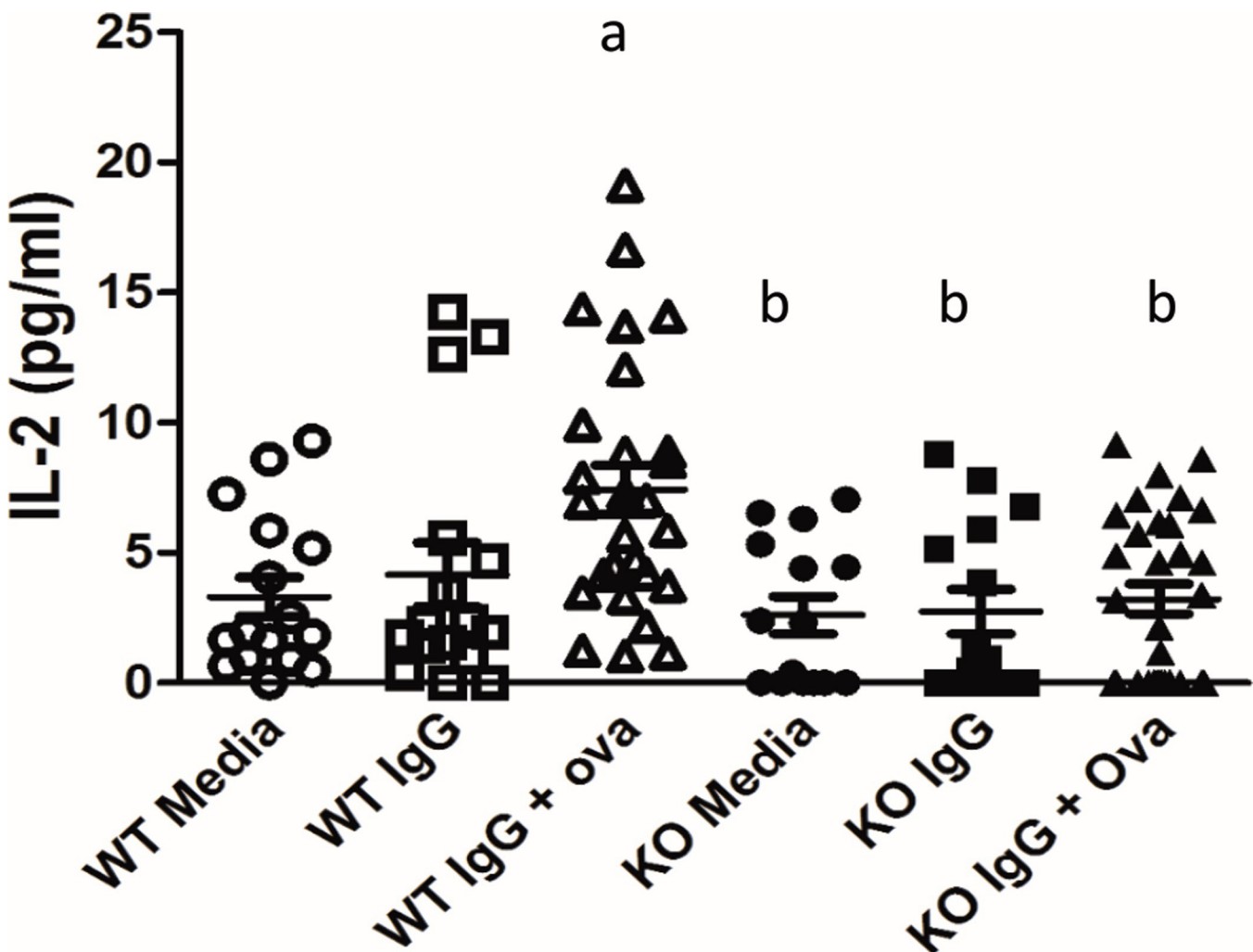

**Fig 2. Antigen presentation in WT and FcRn KO podocytes.** Treatment of podocytes with immune complexes (IgG + NP-Ova) resulted in a significant increase in T cell IL2 production in WT but not FcRn KO podocytes. a, $p < 0.05$ compared to WT + media alone; b, $p = 0.0016$ compared to WT + immune complexes.

## Podocyte-specific KO of FcRn did not alter neutrophil invasion or complement component 3 (C3) deposition after induction of anti-GBM nephritis

In order to determine whether podocyte-specific knockout of FcRn would protect against induction of glomerulonephritis, we generated podocyte-specific FcRn KO (podFcRn KO) mice by crossing podocin-Cre mice with FcRn floxed mice to create FcRn fl/fl:Podocin-Cre/+ mice (Fig 3) as previously described [37]. FcRn floxed mice lacking the Cre transgene (FcRn fl/fl;+/+) served as littermate controls. Podocyte-specific knockout of FcRn did not alter renal histology at 3 months of age (for the anti-GBM model mice were used between 8 and 12 weeks of age; Fig 3A). Since FcRn is known to play an important role in IgG recycling, rabbit IgG

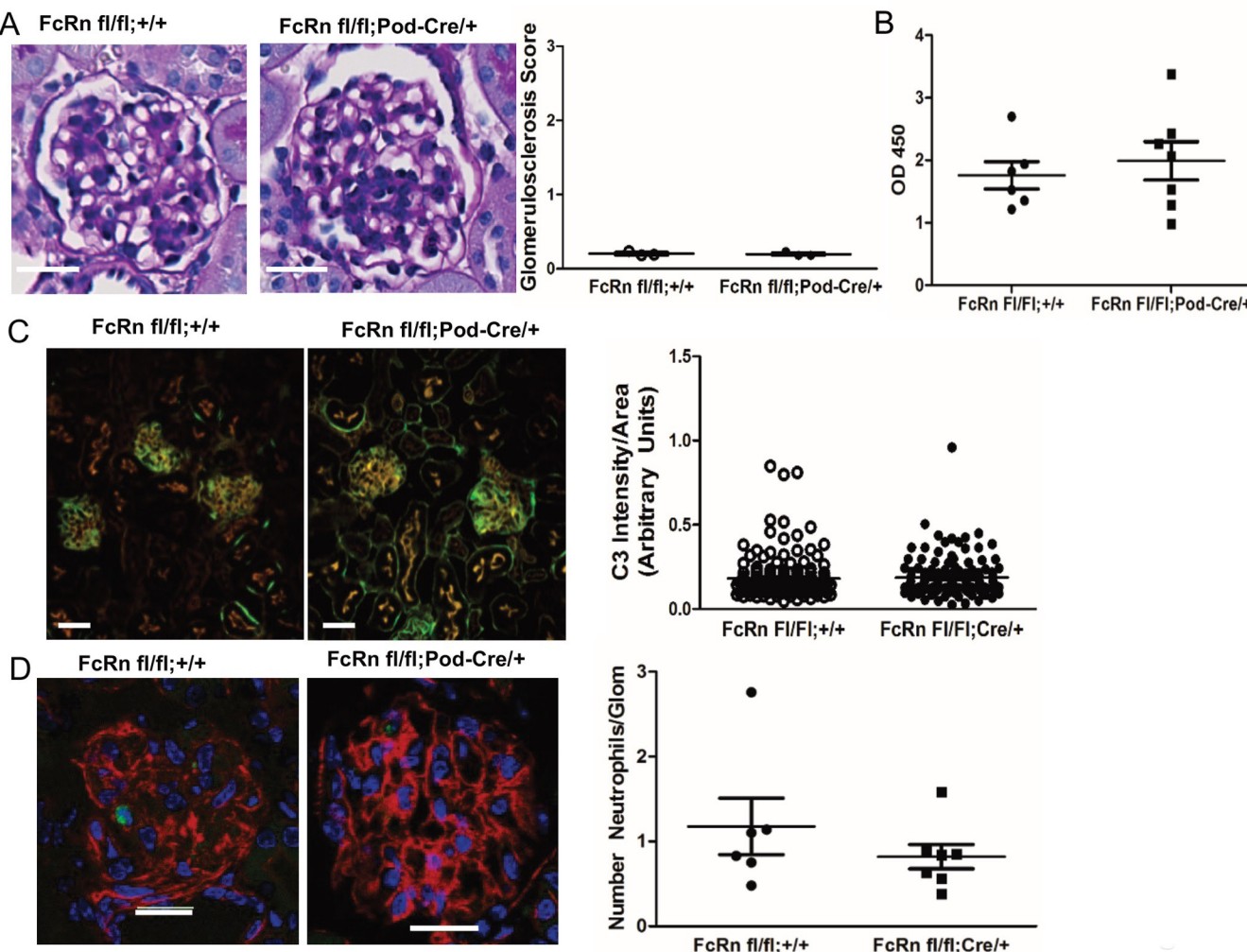

**Fig 3. Characterization of podocyte-specific FcRn knockout mice and induction of anti-GBM nephritis.** *A*. Left Panels: *Representative images of glomeruli from 3 month old* control (FcRn fl/fl;+/+) and podocyte-specific FcRn KO (FcRn fl/fl;Pod-Cre/+). Scale bar 10 μm. Right Panel: minimal glomerulosclerosis at 3 months (n = 3 control and n = 3 KO mice) and no crescents were observed. *B*, Anti-rabbit IgG titers obtained from control and podFcRn KO mice demonstrated no significant difference between the two groups 3 days after the injection of the rabbit anti-mouse GBM antibody. *C*, Left panels: Representative immunofluorescent staining images of podFcRn KO and control mice for C3 (green) and actin (red) 3 days after injections of anti-mouse GBM antibody. Scale bar 20 μm. There was no difference in C3 staining intensity between control and podFcRn KO (right panel; n = 6 control and n = 7 podocyte specific KO mice). *D*, Left panels: Representative images of immunofluorescent staining for Ly6 (green) for neutrophil quantification. Actin is blue. Scale bar 20 μm. Right panel: there was no significant difference in the number of neutrophils per glomerular cross section between the podFcRn KO and controls (numbers of mice same as in *E*).

titers were checked in podFcRn KO mice and controls 3 days after injection of rabbit anti-mouse anti-GBM antibodies to ensure that antibody levels were similar. There was no difference in anti-rabbit IgG titers in controls versus podFcRn KO, indicating that knockout of FcRn in podocytes does not significantly alter IgG metabolism (Fig 3B).

Neutrophil invasion and complement component 3 (C3) deposition occurs in the heterologous phase of the anti-GBM model (within 1–4 days of anti-GBM injection) [30]. To determine if podocyte-specific KO of FcRn had any effect on the early inflammatory response, glomerular neutrophil invasion and C3 deposition were assessed in control and podFcRn KO mice. There was no significant difference in C3 deposition (Fig 3C) or the number of neutrophils per glomerular cross section (Fig 3D) in control or podFcRn KO mice 3 days after injection of anti-GBM antibody, indicating that podocyte-specific KO of FcRn does not attenuate the early inflammatory response.

## Podocyte specific FcRn knockout protects against crescents formation at 8 days after anti-GBM induction

To examine the effects of podocyte-specific knockout of FcRn on the autologous phase of anti-GBM nephritis, markers of renal function and disease severity were examined in podFcRn KO and control mice 8 days after injection of anti-GBM antibodies. At day 8 after induction of anti-GBM nephritis, there was no significant difference in albuminuria, blood urea nitrogen (BUN), glomerulosclerosis scores, glomerular C3 staining, CD4+ T cell or macrophage infiltration, in control versus podFcRn KO (Fig 4A–4F, n = 10 control and n = 10 PodFcRn KO mice). However, PodFcRn KO mice showed a significant decrease in the percentage of glomerular crescents at day 8 compared to control mice (8.5 +/- 2.0% crescents in control vs. 3.4 +/- 1.1% crescents in KO, p = 0.04, Fig 4G). Of note, although the anti-GBM model resulted in significant albuminuria in both control and podFcRn KO mice, the percentage of crescents seen in the control mice was relatively low. These findings are in accord with some other studies of this model in mice [53–56], making this model less ideal to study glomerulonephritis.

## Podocyte specific FcRn knockout protects against crescent formation and glomerulosclerosis at 28 days after NTS induction

To examine the effects of podocyte-specific knockout of FcRn in a disease with a disease with a prolonged autologous phase, we measured markers of renal function and disease severity 28 days after the induction of nephrotoxic serum nephritis (NTS). At 28 days after induction of NTS, there was a significant reduction in albuminuria in the podoFcRn KO mice compared to control (872.9 +/- 65.40 in controls (n = 8) vs 668.7 +/- 66.45 in podFcRn KO mice (n = 7), p = 0.048, Fig 5A) but not blood urea nitrogen (BUN) (75.13 +/- 5.96 in controls (n = 8) vs 67.18+/-6.50 in podoFcRn (n = 6), p = 0.3897, Fig 5B). PodFcRn KO also had significantly lower glomerulosclerosis scores (1.94 +/- 0.24 in controls vs 1.10 +/- 0.24 in KO, p = 0.03, Fig 5C) and percentage of glomerular crescents (31.98 +/- 5.26% in controls vs 6.34 +/- 1.18% in KO, p = 0.0007, Fig 5D) compared to controls.

## Discussion

Glomerulonephritis have long provided therapeutic challenges and efforts are ongoing to understand the mechanisms involved in these diseases. Within the kidney, podocytes have been proposed to act as APCs since they express some of the molecular machinery required for antigen presentation (MHCII, CD80, FcRn) and podocyte-specific KO of MHC II has been shown to attenuate anti-GBM disease [6]. Here we show that in vitro, WT podocytes have the

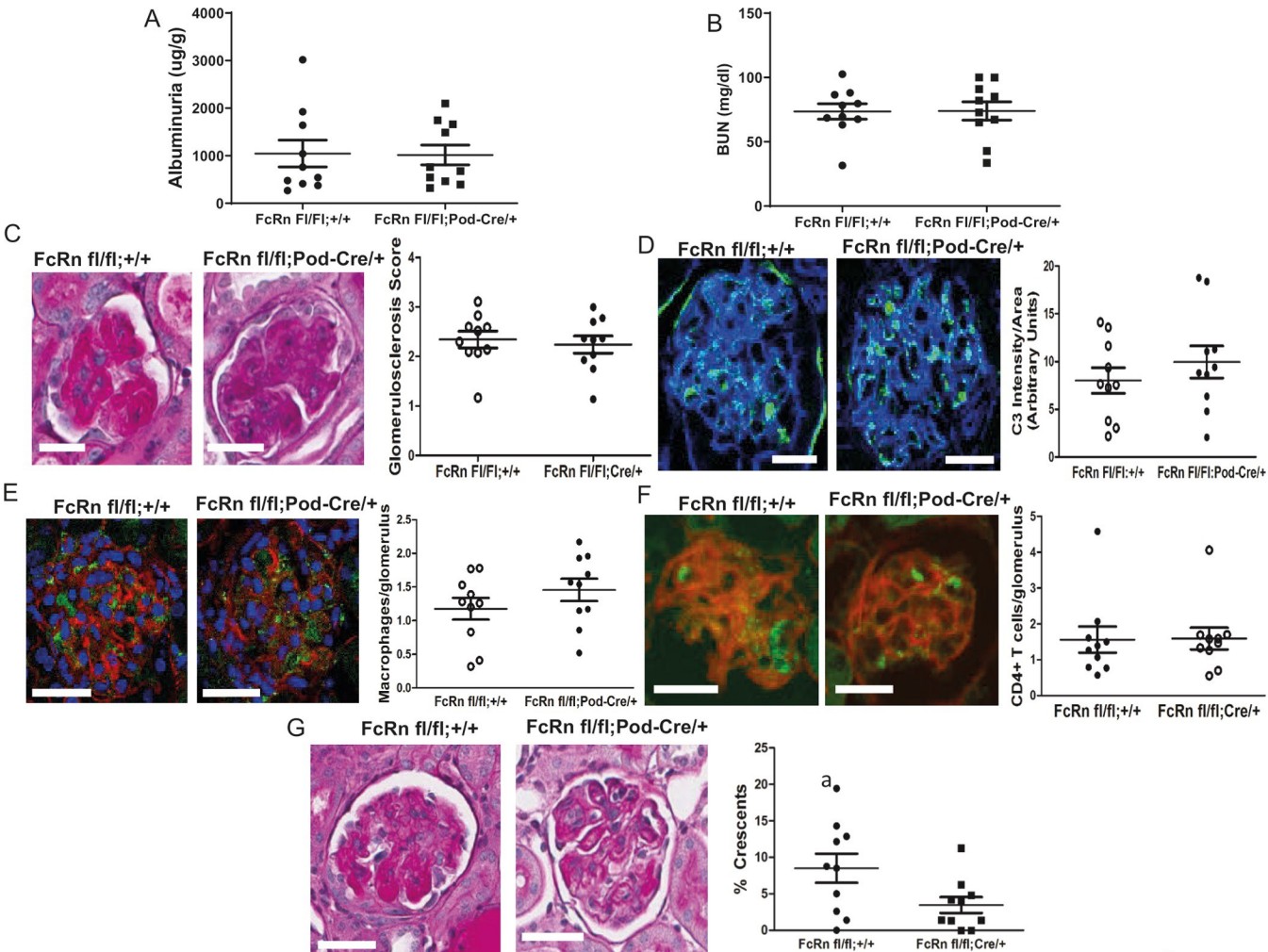

**Fig 4. Podocyte specific KO of FcRn results in a significant decrease in glomerular crescents 8 days after anti-GBM disease induction.** *A*, Urine albumin to creatinine ratios in control and podFcRn KO mice (n = 10 control and n = 10 podFcRnKO mice per group). *B*, BUN in control and podFcRn KO mice. (n = 10 control and n = 10 podFcRnKO mice per group). *C*, Left panel: representative images of histology in control and podFcRn KO and control mice at day 8. Scale bar 20 μm. Right panel: Glomerulosclerosis score in control and podFcRn KO mice (n = 10 control and n = 10 podFcRnKO mice per group). *D*, Left panel: representative images of C3 staining (green) in control and podFcRn KO mice 8 days after anti-GBM disease induction. Actin is stained blue. Scale bar 20 μm. Right panel: Quantification of C3 intensity normalized for glomerular area in control and podFcRn KO mice (n = 10 control and n = 10 podFcRn KO mice). *E*, Left panel: representative images of staining for macrophages (green) in control and podFcRn KO mice 8 days after anti-GBM disease induction. Actin is stained red and nuclei are stained blue. Scale bare 20 μm. Right panel: Quantification of number of macrophages per glomerulus in control and podFcRn Ko mice (n = 10 control and n = 10 podFcRn KO mice). *F*, Left panel: representative images of glomerular staining for CD4+ T cells (green) in control and podFcRn KO mice at day 8 after anti-GBM induction. Actin is stained red. Scale bar 20 μm. Right panel: Quantification of CD4+ cells per glomerulus in control and podFcRn KO mice (n = 10 control and n = 10 podFcRnKO mice per group). *G*, Left panels: representative images of crescents in control mice. Middle Panels: representative images of crescents podFcRn KO mice shown at low and high power. Scale bar 20 μm. Right panel: Percentage of crescents in control and podFcRn KO mice (numbers of mice are same as in C). Podocyte-specific KO of FcRn results in a significant decrease in the number of crescents (a, p = 0.035).

ability to express MHCII and CD80, which are necessary for T cell stimulation, but that WT podocytes have limited ability to activate T cells in vitro. Though inefficient, antigen presentation by podocytes is dependent on FcRn since the KO podocytes were not able to stimulate T cells. Furthermore, we demonstrated that the inability of the KO podocytes to present antigen was not due to abnormal MHCII or costimulatory marker expression since both WT and KO podocytes had similar levels (Fig 1).

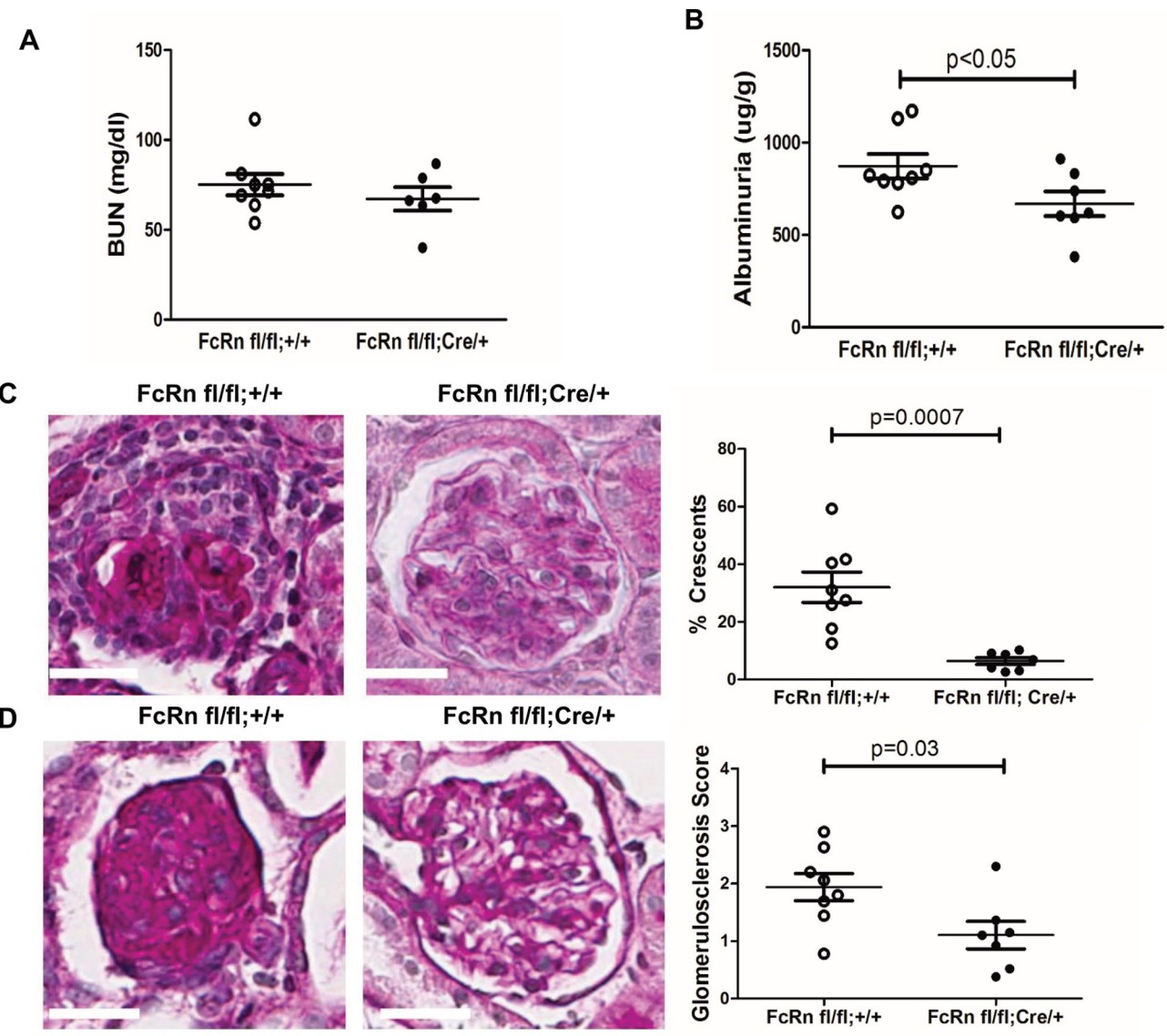

**Fig 5. Podocyte specific KO of FcRn results in a significant decrease in albuminuria, glomerulosclerosis, and glomerular crescents after nephrotoxic serum nephritis induction.** *A*, Urine albumin to creatinine ratios in control were significantly higher than podFcRn KO mice (n = 8 control and n = 7 podFcRnKO mice), p = 0.0480. *B*, BUN in control and podFcRn KO mice were similar (n = 8 control and n = 6 podFcRnKO mice), p = 0.3897. *C*, Left panel: representative images of histology in control mice at day 28. Middle panel: representative image of histology in podFcRn KO at day 28. Scale bar 20 μm. Right panel: Glomerulosclerosis score in control and podFcRn KO mice (n = 8 control and n = 7 podFcRnKO mice per group). Podocyte-specific KO of FcRn results in significant decrease in glomerulosclerosis score compared to control mice (p = 0.03). *D*, Left panel: representative images of crescents in control mice. Middle Panel: representative images of crescents podFcRn KO mice shown. Scale bar 20 μm. Right panel: Percentage of crescents in control and podFcRn KO mice (numbers of mice are same as in C). Podocyte-specific KO of FcRn results in a significant decrease in the number of crescents (p = 0.0007).

Our in vivo data demonstrated that the absence of FcRn in podocytes did not protect against anti-GBM nephritis in the early heterologous stage of the disease. In the autologous phase, podocyte-specific knockout of FcRn resulted in a significant decrease in crescent formation without a significant change in albuminuria, BUN or glomerulosclersosis score. One possibility is that podocyte FcRn is not required for induction of anti-GBM nephritis but may play a role in the resolution phase of the disease. Alternatively, the damage caused by the heterologous phase of the anti-GBM nephritis model may have been too extreme and irreversibly

impacted the albuminuria, BUN, and glomerulosclerosis scores despite a protective effect during the autologous phase provided by knocking out FcRn.

By utilizing the NTS model, we were able to demonstrate that podocyte FcRn plays a significant role in glomerulonephritis diseases that are more autologous in nature. Since there was significantly less albuminuria, glomerulosclerosis, and crescents in the podocyte-specific FcRn knockout mice using the NTS model suggests that the albuminuria and glomerulosclerosis observed in our anti-GBM nephritis model was driven mostly by the heterologous phase. Our nephrotoxic serum nephritis findings also suggest that podocyte FcRn is mainly involved in autologous phase and not during the induction phase.

These findings are significant because it demonstrates a significant difference between the anti-GBM model and the NTS model when investigating the role podocytes play in glomerulonephritis. It further supports the notion that therapies need to be tailored not only to the specific disease but also the particular mechanism underlying each disease process. We found that the anti-GBM model induced a rapid illness in the mice that was often fatal (up to 50% of animals) in a relatively short period of time. The severity of the anti-GBM model may result from the injection with normal rabbit IgG in Freund's complete adjuvant prior to the anti-GBM antibody. This pre-sensitization step may lead to a more severe disease not only due to the immune system being primed to react to the rabbit IgG but also due to the non-specific inflammatory effect from Freund's complete adjuvant.

The NTS model does have a heterologous phase, as reported by the manufacturer, which behaves similarly to the anti-GBM in that it results in IgG and C3 deposition in the glomerulus and proteinuria within a few days. However, contrary to the anti-GBM model, the NTS model had a much lower mortality rate, which could be due to the lack of the pre-sensitization step, thus allowing us to evaluate the disease in the autologous phase. Given the more indolent nature of most glomerulonephritis in humans, NTS seems like the more clinically similar model to study for human application.

Additionally, it has been suggested that podocytes are involved in making glomerular crescents [57–61]. Our observed decrease in glomerular crescents in the podocyte-specific FcRn knockout mice in both models is provocative as it would suggest FcRn is involved in crescent formation. To our knowledge, no prior study has investigated the role of FcRn in crescent formation. The exact mechanisms by which the absence of FcRn prevents crescent formation may have to do with impaired podocyte mobility, alterations in actin dynamics or impaired intracellular signaling. The mechanism underlying how FcRn may affect these pathways is unknown but we have found that knockout of FcRn KO in cultured podocytes alters mobility and actin dynamics [37]. In addition, when comparing the relative quantity of the crescents between the two models, we found significantly fewer glomeruli in the anti-GBM had crescents compared to the NTS model, which is suggestive that crescent formation is more dependent on the autologous phase than the heterologous phase of glomerulonephritis. This information may prove valuable when trying to determine the exact mechanism leading to crescent formation.

A potential area for future study could be aimed at investigating if there is a difference in other models of disease. An example of this might be achieved by breeding our podFcRn KO mouse onto a spontaneous lupus nephritis model which could yield interesting findings. Another area of study would be to create an inducible knockout of podocyte-specific FcRn. This would allow investigation into whether knocking out or blocking FcRn in podocytes after disease onset would modulate the disease course and thus suggest a therapeutic option to treat glomerulonephritis.

In summary, we have shown that induction of anti-GBM nephritis is not dependent on podocyte FcRn but that podocyte-specific knockout of FcRn leads to a significant reduction in

crescent formation. Furthermore, we have shown that podocyte-specific knockout of FcRn can significantly reduce albuminuria, glomerulosclerosis, and crescent formation in the autologous phase of NTS, suggesting podocyte FcRn plays a significant role in disease progression.

## Supporting information

**S1 Fig. IL-2 assay using splenocytes as antigen presenting cells.** There is no T cell production of IL-2 when splenocytes are treated with media or IgG alone but there is a significant increase in IL-2 production when splenocytes are incubated wth immune complexes (IgG + Ova). a, p<0.001.
(TIF)

**S2 Fig. Histologic analysis of control and podocyte FcRn KO mice injected with IgG control and saline.** There was no difference in glomerulosclerosis scores between control mice and podocyte FcRn KO mice injected with saline (sal) nor with control IgG (IgG). n = 2 mice per group. Scale bars in lower left corner are 20μm.
(PDF)

## Author Contributions

**Conceptualization:** Judith Blaine.

**Data curation:** Judith Blaine.

**Formal analysis:** James F. Dylewski, Pantipa Tonsawan, Judith Blaine.

**Funding acquisition:** Judith Blaine.

**Investigation:** James F. Dylewski, Pantipa Tonsawan, Linda Lewis, Judith Blaine.

**Methodology:** James F. Dylewski, Gabriela Garcia, Judith Blaine.

**Writing – original draft:** James F. Dylewski, Judith Blaine.

**Writing – review & editing:** James F. Dylewski, Judith Blaine.

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
