## [Decision Letter · Decision Letter 0]

6 Mar 2020

PONE-D-20-05364

Podocyte-specific knockout of the neonatal Fc receptor (FcRn) results in differential protection depending on the model of immune-mediated kidney disease

PLOS ONE

Dear Dr Dylewski,

Thank you for submitting your manuscript to PLOS ONE. After careful consideration, we feel that it has merit but does not fully meet PLOS ONE’s publication criteria as it currently stands. Therefore, we invite you to submit a revised version of the manuscript that addresses the points raised during the review process.

We would appreciate receiving your revised manuscript by Apr 20 2020 11:59PM. To enhance the reproducibility of your results, we recommend that if applicable you deposit your laboratory protocols in protocols.io, where a protocol can be assigned its own identifier (DOI) such that it can be cited independently in the future. For instructions see: http://journals.plos.org/plosone/s/submission-guidelines#loc-laboratory-protocols

We look forward to receiving your revised manuscript.

Kind regards,

Zhanjun Jia

Academic Editor

PLOS ONE

Additional Editor Comments (if provided):

Please address the concerns from the experts.

Journal Requirements:

2. To comply with PLOS ONE submissions requirements, please provide method(s) of sacrifice/euthanasia in the Methods section of your manuscript.

3. As part of your revision, please complete and submit a copy of the ARRIVE Guidelines checklist, a document that aims to improve experimental reporting and reproducibility of animal studies for purposes of post-publication data analysis and reproducibility: https://www.nc3rs.org.uk/arrive-guidelines. Please include your completed checklist as a Supporting Information file. Note that if your paper is accepted for publication, this checklist will be published as part of your article.

"NO"

Please provide an amended Funding Statement that declares *all* the funding or sources of support received during this specific study (whether external or internal to your organization) as detailed online in our guide for authors at http://journals.plos.org/plosone/s/submit-now.  Please state what role the funders took in the study.  If any authors received a salary from any of your funders, please state which authors and which funder. If the funders had no role, please state: "The funders had no role in study design, data collection and analysis, decision to publish, or preparation of the manuscript."

Reviewers' comments:

Reviewer's Responses to Questions

**Comments to the Author**

1. Is the manuscript technically sound, and do the data support the conclusions?

Reviewer #1: Yes

Reviewer #2: Partly

2. Has the statistical analysis been performed appropriately and rigorously? 

Reviewer #1: Yes

Reviewer #2: Yes

3. Have the authors made all data underlying the findings in their manuscript fully available?

Reviewer #1: Yes

Reviewer #2: Yes

4. Is the manuscript presented in an intelligible fashion and written in standard English?

Reviewer #1: Yes

Reviewer #2: Yes

5. Review Comments to the Author

Reviewer #1: This is an interesting and significant research work. The authors demonstrated that induction of anti-GBM nephritis is not dependent on podocyte FcRn and lack of podocyte FcRn was protective in immune mediated kidney disease that is dependent on an autologous phase. Their findings suggested that specific therapies should be tailored according to the specific disease and the certain stage. The experiments were performed by utilizing two different mouse models of immune-mediated nephritis, the methodologies used were adequate.

Suggestions:

1. Since authors mentioned in the discussion that podocytes are involved in making glomerular crescents and FcRn deficiency alters mobility and actin dynamics in cultured podocytes, it seems logical to check the situation of podocyte injury in vivo in their own FcRn KO mice with or without anti-GBM antibody treatment.

2. The NTS model also has a heterologous phase although within a few days, since authors propose podocyte FcRn as a renoprotective factor mainly involved in autologous phase but not the heterologous phase, it is better to examine the effects of podocyte-specific knockout of FcRn in the heterologous phase of NTS model.

3. The Fig 4 lacks the panel H.

4. Please explain why the control groups were missed in the in vivo experiments (anti-GBM nephritis and NTS neohritis models).

5. At the beginning of this muscript, authors mentioned that immune-mediated nephritis is the third leading cause of end stage kidney disease in the United States (1). While in the Ref 1, authors seems not mention the third leading cause of end stage kidney disease in US. Could the authors provide the detail of its source?

Reviewer #2: In the manuscript entitled as “Podocyte-specific knockout of the neonatal Fc receptor (FcRn) results in differential protection depending on the model of immune-mediated kidney disease”, the authors first proved that podocyte could function as antigen presenting cells. FcRn was implicated in this process by using FcRn KO podocytes. Results in the NTS mice model demonstrated that podocyte-specific knockout of FcRn reduced albuminuria, glomerulosclerosis, and crescent formation. They fairly interpreted the data and clearly presented the results in the manuscript. The quality of the manuscript could be further improved once the following issues had been addressed.

Major concerns:

1. The quality of the images should be improved.

2. Please provide profile of the KO podocyte, e.g. the knockout efficiency, the cellular characteristics of the podocytes, in Figure 1. Does the FcRn knockout affect the morphology of the podocytes?

3. Please prove that FcRn was specifically knocked out in the podocyte in the animal model.

4. The authors collected urine for the analysis of albuminuria. How was the urine collected? Was it 12h or 24h urine?

5. Other than what described in Figure 3, is there any abnormality in the renal function in the FcRn fl/fl;Pod-Cre mice?

6. Normal controls that were not injected with anti-GBM antibody should be included in Figure 3 &4 to prove the establishment of model.

7. The quality of the immunofluorescent staining shown in Figure 3D should be improved, as the green fluorescence with the similar intensity as the positive staining also present in the tubular.

Minor concerns:

1. Was the person who performed the histological scoring of glomeruli blinded to the grouping?

2. Please describe the successful rate and mortality rate of anti-GBM nephritis model and nephrotoxic serum nephritis models in the Methods section. How many mice per group were used at the start of modeling?

3. In the Methods for the immunofluorescence for fixed cells, please name the type of normal serum used for blocking.

4. In Figure 1, the authors found that IFNγ-stimulation induced significant increase of MHCII in the podocytes. Please specify the control used for comparison.

5. Please discuss the results in Figure 1.

6. The numbering of the images in Figure 3 was not consistent with the figure legend.

7. What is the unit of the Y axis in Figure 3B?

8. Please counter-stain the nuclei in the Figure 3C.

6. PLOS authors have the option to publish the peer review history of their article (what does this mean?). If published, this will include your full peer review and any attached files.

Reviewer #1: No

Reviewer #2: No

---

## [Author Response · Author response to Decision Letter 0]

3 Dec 2020

We appreciate the reviewers’ detailed and thoughtful comments that have helped to improve this manuscript. Below is our response to the reviewers:

Reviewer #1:

Since authors mentioned in the discussion that podocytes are involved in making

glomerular crescents and FcRn deficiency alters mobility and actin dynamics in

cultured podocytes, it seems logical to check the situation of podocyte injury in vivo

in their own FcRn KO mice with or without anti-GBM antibody treatment.

We have provided histologic analysis of control and podocyte specific FcRn KO mice injected with saline as well as a non-specific rabbit IgG control in supplemental figure 2. We observed no difference in glomerulosclerosis scores between the control and podocyte specific FcRn KO mice injected with saline or rabbit IgG. 

The NTS model also has a heterologous phase although within a few days, since

authors propose podocyte FcRn as a renoprotective factor mainly involved in

autologous phase but not the heterologous phase, it is better to examine the effects

of podocyte-specific knockout of FcRn in the heterologous phase of NTS model.

Our aim in this paper was to determine the ability of podocytes to act as antigen presenting cells and whether targeting the podocytes ability to present antigen could attenuate the adaptive immune response leading to glomerulonephritis. The heterologous phase of the NTS consists of the innate immune response which should not be dependent on antigen presentation by podocytes. As such, we feel the study of the podocyte FcRn on the innate immune response extends beyond the scope of this paper and should be investigated separately. 

The Fig 4 lacks the panel H.

This was a labeling error and it has been corrected. 

Please explain why the control groups were missed in the in vivo experiments

(anti-GBM nephritis and NTS nephritis models).

The in vivo experiments were performed in podocyte specific FcRn knockout mice and control mice (FcRn fl/fl;+/+) which lacked the cre expression and thus had normal FcRn expression. All analysis compared the control mice (FcRn fl/fl;+/+) and the podocyte specific FcRn knockout mice (podoFcRn fl/fl;Cre/+) to determine the relative effect podocyte FcRn had on disease severity. 

“Sham” controls of intravenous saline and non-specific IgG were performed and now included in Supplemental Figure 2.

At the beginning of this manuscript, authors mentioned that immune-mediated

nephritis is the third leading cause of end stage kidney disease in the United States

(1). While in the Ref 1, authors seems not mention the third leading cause of end

stage kidney disease in US. Could the authors provide the detail of its source?

The United States Renal Data Systems (USRDS), lists glomerulonephritis as the third leading cause for End Stage Kidney disease in the United States. We have changed our wording of “immune mediated kidney disease” to “glomerulonephritis” in order to be more specific to the type of diseases we are investigating. 

Reviewer #2

Major concerns:

The quality of the images should be improved.

Representative images using the highest resolution of the images available were included in this resubmission. Furthermore, the PDFs containing these images are now saved as ‘Press Quality’ to ensure the resolution is preserved during the uploading process. Some images are more pixelated because these images were obtained at a lower power magnification that resulted in the loss of resolution when zoomed in to focus on individual glomeruli. 

Please provide profile of the KO podocyte, e.g. the knockout efficiency, the

cellular characteristics of the podocytes, in Figure 1. Does the FcRn knockout affect

the morphology of the podocytes?

WT and KO podocytes were characterized in the following paper: Differential trafficking of albumin and IgG facilitated by the neonatal Fc receptor in podocytes in vitro and in vivo Dylewski et al., PLoS One 2019, PMCID: PMC6392300. KO podocytes have similar morphology to KO podocytes but do have differences in the actin cytoskeleton (Knockout of the neonatal Fc receptor in cultured podocytes alters IL-6 signaling and the actin cytoskeleton. Tonsawan et al., AJP Cell Physiol, 2019 PMCID: PMC6879880).

Please prove that FcRn was specifically knocked out in the podocyte in the animal

model.

Podocyte-specific KO mice were characterized in: Differential trafficking of albumin and IgG facilitated by the neonatal Fc receptor in podocytes in vitro and in vivo Dylewski et al., PLoS One 2019, PMCID: PMC6392300. Additional characterization as well as details on genotyping and generation of these mice are described in Generating a Podocyte-Specific Neonatal Fc Receptor (FcRn) Knockout Mouse. Blaine J. Methods Molecular Biology, Vol. 2224, ShreeRam Singh et al. (Eds): Mouse Genetics, Springer Nature, 2020, accepted. In this chapter we show that staining for FcRn in vivo is absent in the glomerular locations in which podocytes are found in podocyte-specfic FcRn KO mice but is present in the endothelium. We also demonstrate that staining for FcRn is minimal in podocytes isolated from podocyte-specific FcRn KO mice compared to podocytes isolated from control mice.

The authors collected urine for the analysis of albuminuria. How was the urine

collected? Was it 12h or 24h urine?

The urine was a spot (random collection)

Other than what described in Figure 3, is there any abnormality in the renal

function in the FcRn fl/fl;Pod-Cre mice?

 In our previous paper, Differential trafficking of albumin and IgG facilitated by the neonatal Fc receptor in podocytes in vitro and in vivo Dylewski et al., PLoS One 2019, PMCID: PMC6392300, we show that 3 month old podocyte-specific FcRn KO mice have normal levels of serum albumin, similar albuminuria and similar GFRs compared to control mice. In the experiments described in this paper we used mice that were 10 – 12 weeks of age.

Normal controls that were not injected with anti-GBM antibody should be included

in Figure 3 &4 to prove the establishment of model.

The in vivo experiments were performed in podocyte specific FcRn knockout mice and control mice (FcRn fl/fl;+/+) which lacked the cre expression and thus had normal FcRn expression. All analysis compared the control mice (FcRn fl/fl;+/+) and the podocyte specific FcRn knockout mice (podoFcRn fl/fl;Cre/+) to determine the relative effect podocyte FcRn had on disease severity. 

“Sham” controls of intravenous saline and non-specific IgG were performed and now included in Supplemental Figure 2.

The quality of the immunofluorescent staining shown in Figure 3D should be

improved, as the green fluorescence with the similar intensity as the positive

staining also present in the tubular.

The staining in Figure 3D has been redone.

Minor concerns:

1. Was the person who performed the histological scoring of glomeruli blinded to the

grouping?

Yes, all histologic analysis was performed in a blinded fashion.

2. Please describe the successful rate and mortality rate of anti-GBM nephritis

model and nephrotoxic serum nephritis models in the Methods section. How many

mice per group were used at the start of modeling?

For the anti-GBM mice mortality was as follows: one control and two podocyte-specific FcRn Ko mice died during the experiment. Starting numbers were 11 control and 12 podocyte-specific FcRn KO mice.

For the NTS experiments, one control and one podocyte-specific FcRn KO mouse died so starting numbers were as follows: 9 control and 8 podocyte-specific FcRn KO mice. This information has been added to the Methods section.

3. In the Methods for the immunofluorescence for fixed cells, please name the type

of normal serum used for blocking.

This section was included in error. In this manuscript there is no immunofluorescence on fixed cells so this section of the Methods has been removed.

4. In Figure 1, the authors found that IFNγ-stimulation induced significant increase

of MHCII in the podocytes. Please specify the control used for comparison.

The control used was podocytes treated with regular media without any IFN�.

5. Please discuss the results in Figure 1.

It is established that cells, including podocytes, can express MHCII after exposure to INF�. Since FcRn is involved in antigen loading on MHCII, the experiments in Figure 1 were performed to ensure that knockout of FcRn did not alter the podocytes ability to express MHCII and co-stimulatory markers and thus impair the FcRn KO podocytes ability to present antigen. This has been added to the discussion section.

6. The numbering of the images in Figure 3 was not consistent with the figure

legend.

The numbering has been changed to ensure they are correct and aligned with what is described in the text.

7. What is the unit of the Y axis in Figure 3B?

The units for the Y axis in Figure 3B is OD450 and have been added to the figure.

8. Please counter-stain the nuclei in the Figure 3C.

Figure 3C is demonstrating the presence of C3 within the kidney. Since C3 deposition is not specific to the location of cells, nuclei staining was not performed in this portion of the analysis. Staining for actin was performed to identify structural components within the kidney.

---

## [Decision Letter · Decision Letter 1]

9 Dec 2020

Podocyte-specific knockout of the neonatal Fc receptor (FcRn) results in differential protection depending on the model of glomerulonephritis

PONE-D-20-05364R1

Dear Dr. Dylewski,

We’re pleased to inform you that your manuscript has been judged scientifically suitable for publication and will be formally accepted for publication once it meets all outstanding technical requirements.

Kind regards,

Zhanjun Jia

Academic Editor

PLOS ONE

Additional Editor Comments (optional):

Reviewers' comments:

Reviewer's Responses to Questions

**Comments to the Author**

1. If the authors have adequately addressed your comments raised in a previous round of review and you feel that this manuscript is now acceptable for publication, you may indicate that here to bypass the “Comments to the Author” section, enter your conflict of interest statement in the “Confidential to Editor” section, and submit your "Accept" recommendation.

Reviewer #1: All comments have been addressed

Reviewer #2: All comments have been addressed

2. Is the manuscript technically sound, and do the data support the conclusions?

Reviewer #1: Yes

Reviewer #2: (No Response)

3. Has the statistical analysis been performed appropriately and rigorously? 

Reviewer #1: Yes

Reviewer #2: (No Response)

4. Have the authors made all data underlying the findings in their manuscript fully available?

Reviewer #1: Yes

Reviewer #2: (No Response)

5. Is the manuscript presented in an intelligible fashion and written in standard English?

Reviewer #1: Yes

Reviewer #2: (No Response)

6. Review Comments to the Author

Reviewer #1: (No Response)

Reviewer #2: (No Response)

7. PLOS authors have the option to publish the peer review history of their article (what does this mean?). If published, this will include your full peer review and any attached files.

Reviewer #1: No

Reviewer #2: No

---

## [Editor Report · Acceptance letter]

14 Dec 2020

PONE-D-20-05364R1 

Podocyte-specific knockout of the neonatal Fc receptor (FcRn) results in differential protection depending on the model of glomerulonephritis. 

Dear Dr. Dylewski:

I'm pleased to inform you that your manuscript has been deemed suitable for publication in PLOS ONE. Congratulations! Your manuscript is now with our production department. 

Kind regards, 

on behalf of

Dr. Zhanjun Jia 

Academic Editor

PLOS ONE